# Dynamic and asymmetric colloidal molecules

Huang Fang [1,10], Qiong Gao [1,10], Yujie Rong[1,10], Yanshuang Chen[1], Jiping Huang [1,2], Hua Tong [3], Zhihong Nie [4,5] ✉, Hajime Tanaka [6,7] ✉, Wei Li [5,8] ✉ & Peng Tan [1,9] ✉

"Colloidal molecules" represent artificial colloidal clusters replicating the geometries of molecules and exhibiting flexibility and fluctuations similar to macromolecules and proteins. Their dynamic and anisotropic characters make them unique and indispensable building blocks for creating hierarchically organized superstructures. Despite the progress in synthesizing and assembling colloidal molecules, unveiling their dynamic characters is challenging in experiments. Here, we employ real-time three-dimensional imaging and simulations to reveal dynamic colloidal molecule structures in micrometre-sized colloidal-emulsion models with tunable electrostatic interactions. Our findings reveal that colloidal molecules' dynamic structures are inherently asymmetric, with angular symmetry emerging through continuous ordering from a liquid-like configuration. We further develop an effective method to guide the ordering of colloidal molecules towards a desired structure by dynamically adjusting the ionic strength in the solvent during the ordering process. We validate this method using molecular dynamics simulations and propose a practical protocol for its experimental implementation. Our research contributes to a clearer physical understanding of dynamic colloidal molecules and offers potential solutions to the complexities inherent in their formation process.

"Colloidal molecules" (CMs) refer to clusters of colloidal particles exhibiting specific local angular arrangements that replicate the geometries of molecules[1–6]. These inherent and stable angular orders significantly enhance CMs' self-assembly capacities. CMs stand out as essential and distinctive building blocks, facilitating the creation of predetermined hierarchical superstructures[7–9].

Previous studies have utilised solutions to the Thomson problem, which addresses the arrangement of $N$ electrons on a sphere, to achieve specific angular orders and symmetry in CMs[10–12]. Building on this, significant efforts have focused on synthesizing CMs with fixed satellites and precisely controlled bond angles[13–15]. These fixed CMs are crucial for assembling complex crystalline structures that require directional bonding[16–18].

Moreover, dynamic CMs, featuring configurational flexibility and fluctuations, have garnered increasing attention from physicists and materials scientists[19–21]. Unlike CMs with fixed satellites, dynamic CMs

[1]Department of Physics and State Key Laboratory of Surface Physics, Fudan University, Shanghai 200438, P. R. China. [2]Key Laboratory of Micro and Nano Photonic Structures (MOE), Fudan University, Shanghai 200438, P. R. China. [3]Department of Physics, University of Science and Technology of China, Hefei 230026, P. R. China. [4]Department of Macromolecular Science, Fudan University, Shanghai 200438, P. R. China. [5]State Key Laboratory of Molecular Engineering of Polymers, Fudan University, Shanghai 200438, P. R. China. [6]Research Center for Advanced Science and Technology, University of Tokyo, 4-6-1 Komaba, Meguro-ku, Tokyo 153-8904, Japan. [7]Department of Fundamental Engineering, Institute of Industrial Science, University of Tokyo, 4-6-1 Komaba, Meguro-ku, Tokyo 153-8505, Japan. [8]Department of Chemistry and Laboratory of Advanced Materials, Fudan University, Shanghai 200438, P. R. China. [9]Institute for Nanoelectronic Devices and Quantum Computing, Fudan University, Shanghai 200438, P. R. China. [10]These authors contributed equally: Huang Fang, Qiong Gao, Yujie Rong. ✉e-mail: znie@fudan.edu.cn; tanaka@iis.u-tokyo.ac.jp; weilichem@fudan.edu.cn; tanpeng@fudan.edu.cn

more closely resemble biological structures, such as macromolecules and proteins, where significant configurational fluctuations occur. This flexibility enables them to bind more effectively with irregularly arranged or fluctuating binding sites, leading to more efficient reactions[22–24]. Designing stable and flexible CMs has been achieved through patchy colloids with directional bonding and the application of mobile DNA linkers[25–29]. Recent studies suggest that dynamic CMs tune their angular orders through a delicate interplay between entropy and electrostatic interactions[30]. Managing these configurational fluctuations, as seen in biological systems, opens new possibilities for the creation of amorphous and polymorphic materials[31,32].

Despite progress in synthesising and assembling CMs, our understanding of their physical characteristics still needs to be improved. For instance, CMs are usually considered symmetric, but how do we adjust the asymmetry of dynamic CMs? On the kinetic side, achieving a CM with a specific angular order is challenging due to multiple and sequential reaction processes among colloidal particles[30]. However, experimental observation, characterisation, and tuning of CMs' dynamic structures pose challenges due to both the fast internal and global translational-rotational motion of CMs in the solvent[19]. These challenges impede a clear understanding of the dynamic features of CMs.

Here, we design a colloidal-emulsion model system that forms micrometre-sized dynamic CMs with tunable electrostatic interactions[33,34]. By adjusting the ionic strength, we can effectively control the range and the magnitude of electrostatic repulsion between charged satellite particles. This system enables three-dimensional imaging of CMs' dynamic structures and formation kinetics under confocal microscopy. Through a combination of experiments and numerical simulations, we discover the intrinsic asymmetry in CMs' dynamic structures. Their angular symmetry evolves continuously from a liquid-like configuration, distinct from a sharp first-order transition. We further propose a method to guide CMs' ordering kinetics toward the desired target structure by dynamically adjusting the ionic strength in the solvent. This approach offers a solution to the complexities inherent in the formation process.

## Results and Discussion
### Model system design
Our model CMs consist of satellite PMMA colloidal particles (negatively charged, diameter $\sigma_s \approx 1.2$ µm, suspended in a density and refrective-index matching cyclohexyl bromide (CHB)-*cis*-decalin solvent) and central emulsion droplets (negatively charged, diameter $\sigma_c$ ranging between 0.8 µm and 1.5 µm, water-glycerol droplet in a CHB-decalin solvent, with sodium dodecyl sulfate (SDS) serving as a surfactant), as illustrated in Fig. 1a. Although gravity could potentially influence the configuration of CMs, we neglect its effect in our model due to the minimal density difference among the PMMA particles, the CHB-decalin solvent, and the emulsions (see Methods for a detailed discussion). The repulsion between satellite particles is modelled by a hard-core Yukawa pair potential, $U_{pp}(r) = \frac{Q^2}{\sigma_s(1+\frac{\kappa\sigma_s}{2})^2\epsilon_s} \exp[-\kappa\sigma_s(r/\sigma_s-1)]/(r/\sigma_s)$ for $r > \sigma_s$, with $\kappa^{-1}$ representing the screening length, and $\frac{Q^2}{\sigma_s(1+\frac{\kappa\sigma_s}{2})^2\epsilon_s}$, denoted as $A_r$, being a surface potential parameter. Here, $Q$ denotes the surface charge of the colloidal particle, and $\epsilon_s$ represents the dielectric constant of the solvent. The interaction between particles and the droplet involves a long-range repulsion and a short-range attraction due to charge-image charge attraction and hydrophobic forces[33,34]. Particles can move freely on the emulsion droplet surface. CMs are formed by mixing colloidal suspensions with emulsions (see Methods). Tetrabutylammonium bromide (TBAB) salt is introduced into the solvent to modify $\kappa^{-1}$ and $Q$. We primarily conducted experiments at two salt concentrations: 1 µM and 5 µM. As the TBAB concentration increases, the surface charge $Q$ rises, the Debye screening length $\kappa^{-1}$ decreases, and the surface potential parameter $A_r$ slightly decreases (see Methods for calculation details). By

adjusting the stoichiometry ratio to approximately 200:1 between particles and emulsion droplets, we effectively inhibit the formation of large aggregations.

### Angular order and dynamic fluctuations of CMs
Our experiments yielded dynamic colloidal molecules (N-CMs) with a satellite particle number $N$ ranging from 2 to 6 (Fig. 1b). At 5 µM TBAB ($\kappa\sigma_s \approx 4.5$), observed configurations include 2-CMs in a V shape, 3-CMs forming a triangle, 4-CMs adopting a tetrahedral structure, 5-CMs exhibiting both triangular bipyramids and pyramid shapes, and 6-CMs forming an octahedral configuration. Importantly, these N-CMs exhibit stability and flexibility, displaying configurational fluctuations around an equilibrium state characterised by angular orders. Real-time trajectories of single CMs obtained in experiments (Fig. 1c and Movies S1-5) reveal satellite particles undergoing temporal fluctuations and unrestricted movement around the core.

Utilising the Kabsch algorithm for optimal satellite alignment through rotation[35,36] (see Methods), we experimentally obtained dynamic structures, including both the average and the fluctuation, by examining multiple CMs with similar core sizes. We emphasize that the system reached equilibrium by the end of our experiment, as the temporal distribution of a single CM was statistically similar to the ensemble distribution obtained from multiple CMs. Interestingly, the average configurations of satellite particles display asymmetry, deviating from the standard symmetric configurations predicted by solutions to the Thomson problem. For instance, the angles in V-shaped configurations may not be the ideal 180 degrees, and triangular configurations are not always equilateral. Intriguingly, despite the triangular bipyramid being the ideal configuration, we also observed the presence of pyramid configurations for 5-CMs.

Figure 1 d shows the distribution of CMs' configurations with coordination numbers from 2 to 6, derived from snapshots of over 50 independent CMs at equilibrium (1 µM TBAB, $\kappa\sigma_s \approx 2.0$). For 2-CMs, the angle distribution of the V shape centres around 118 degrees, with a fluctuation of approximately 30 degrees (see also Movie S1). For 3-CMs, the top view reveals an acute triangle as the average configuration instead of an equilateral triangle (Movie S2). For 4-CMs, the tetrahedron is irregular, with two faces nearly perpendicular to each other (Movie S3). For 5-CMs, we classify the entire population into two categories based on their proximity to either the triangular bipyramid or the pyramid (see Methods). Approximately 20% of the observed configurations are closer to the triangular bipyramid, while the majority are closer to the pyramid (Movie S4). The average configuration within each category closely resembles their respective reference. For 6-CMs, four satellites in the middle remain roughly on the same plane below the equator, yet forming distorted pyramids in both the upper and bottom halves (Movie S5).

Our experimental findings apply universally to CMs assembled by charged colloids, e.g., colloidal particles grafted by various types of functional groups, which act as satellites in aqueous solutions or organic solutions[5,6,29]. In these systems, key experimental parameters, such as ion concentration in the solvent, zeta potential (surface potential), and the size of colloidal particles, play a critical role in quantifying the charged interactions. To systematically and comprehensively understand dynamic CMs' asymmetry and to assess the universality, we perform canonical Monte Carlo simulations using $\kappa\sigma_s$, $A_r$, and $\sigma_c/\sigma_s$ as control parameters (see Methods).

Interestingly, the dynamic CMs' asymmetry appears to be universal. By manipulating $\kappa\sigma_s$ and $A_r$, we can create CMs with specific dynamic structures. Figure 2a provides guidance for designing asymmetric 2-CMs with an average bond angle, $\langle\theta\rangle$, ranging from 100° to 150° ($\sigma_c/\sigma_s = 0.95$). Similarly, for 3-CMs, we can adjust their asymmetry, quantified by the solid angle, $\Omega$ (the area of the segment of a unit sphere, $\Omega = 2\pi$ in steradian for symmetric 3-CMs), as illustrated in Fig. 2b. Note that the standard deviation of $\theta$ (2-CMs) and $\Omega$ (3-CMs)

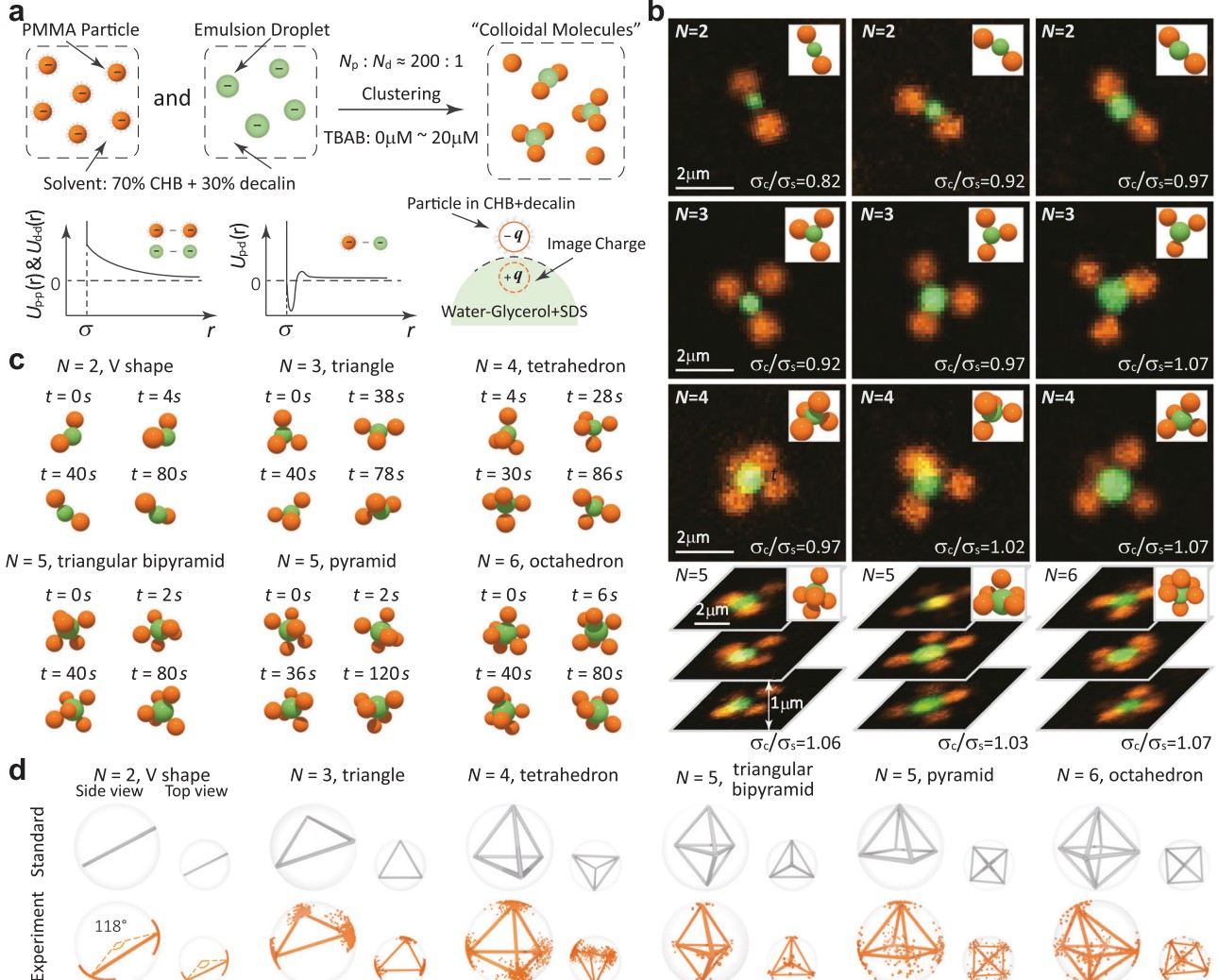

**Fig. 1 | Dynamic structures of colloidal molecules observed in experiments.** **a** Schematic representation of the experimental system featuring tunable electrostatic interactions. **b** Confocal microscopy images and reconstructed three-dimensional configurations of $N$-CMs with varying $\sigma_c$ (5 μM TBAB). The scanning speed along $Z$ direction is 7 μm/s (250 nm per slice). **c** Illustration of the temporal fluctuations of a single CM in experiments, portrayed through real-time trajectories. See also Movies S1-5. **d** Configurational fluctuations of $N$-CMs (bottom raw, 1 μM TBAB) in comparison with the corresponding standard Thomson's solution (top row) for each $N$. Side and top views are presented, with dots indicating satellite positions after rotation for maximal superimposition. We classify 5-CMs based on their proximity to either a triangular bipyramid or a pyramid configuration.

decreases consistently with a reduction in CMs' asymmetry (Figs. S1a and b).

For $N \geq 4$ CMs, we quantify the asymmetric and fluctuated structure profile by using two parameters, $\delta$, representing the deviation between the average (asymmetric) and standard (symmetric) configuration, and $\Delta$, indicating the magnitude of fluctuations around the average configuration (see Methods for definitions). Figure 1c and Figs. S1c-e display the diagram modifying $\delta$ and $\Delta$ for 4-CMs and 6-CMs, respectively. Notably, all CMs tend to converge to solutions of Thomson problems as $\kappa\sigma_s$ and $\sigma_c$ decrease and $A_r$ increases. This behaviour is expected as inter-satellite repulsive interactions become sufficiently long-range to stabilise regular configurations corresponding to solutions to the Thomson problem.

We compare the experimental configurations at different TBAB concentrations with the corresponding simulations (see Methods for technical details on the comparison). As the TBAB concentration decreases from 5 μM ($\kappa\sigma_s = 4.5$, $A_r = 4.7$ $k_BT$) to 1 μM ($\kappa\sigma_s = 1.9$, $A_r = 6.5$ $k_BT$), the configurations of 2, 3, and 4-CMs observed in experiments become more regular, though they still exhibit significant asymmetry. This trend is in full agreement with the simulation

predictions. Notably, unlike in simulations, where $\kappa\sigma$ and $A_r$ can be varied independently, in experiments both the Debye screening length and the particle surface charge are functions of the salt concentration. The tunability of the asymmetry strongly depends on the physicochemical properties of the system, including the particle material, solvent choice, and type of salt solute.

Nevertheless, the optimal ranges for asymmetric CM assembly can be universally adapted to other experimental systems. From our simulations, we have identified that to achieve asymmetric CMs, the range of $\kappa\sigma_s$ should ideally be between 1 and 20, while the surface potential should range from 10 to 100 $k_BT$. For our system with 1-μm diameter particles, the optimal $\zeta$ potential is between 10 and 20 mV, with the TBAB concentration around 1 μM. In an aqueous solvent (where the relative dielectric constant $\epsilon_r \approx 80$), given that $A_r \propto \epsilon_r\sigma_s\zeta^2$ and $\kappa \propto \sqrt{n_{ion}/\epsilon_r}$[37], achieving similar conditions would require adjustments such as using a smaller particle size ($\sigma_s$ between 200 and 500 nm), lower $\zeta$ potential (5 to 10 mV), and an increase in ionic strength ($n_{ion}$ between 10 and 100 μM).

The emergence of asymmetric structures can be elucidated through the interplay of entropy and potential energy. Take, for

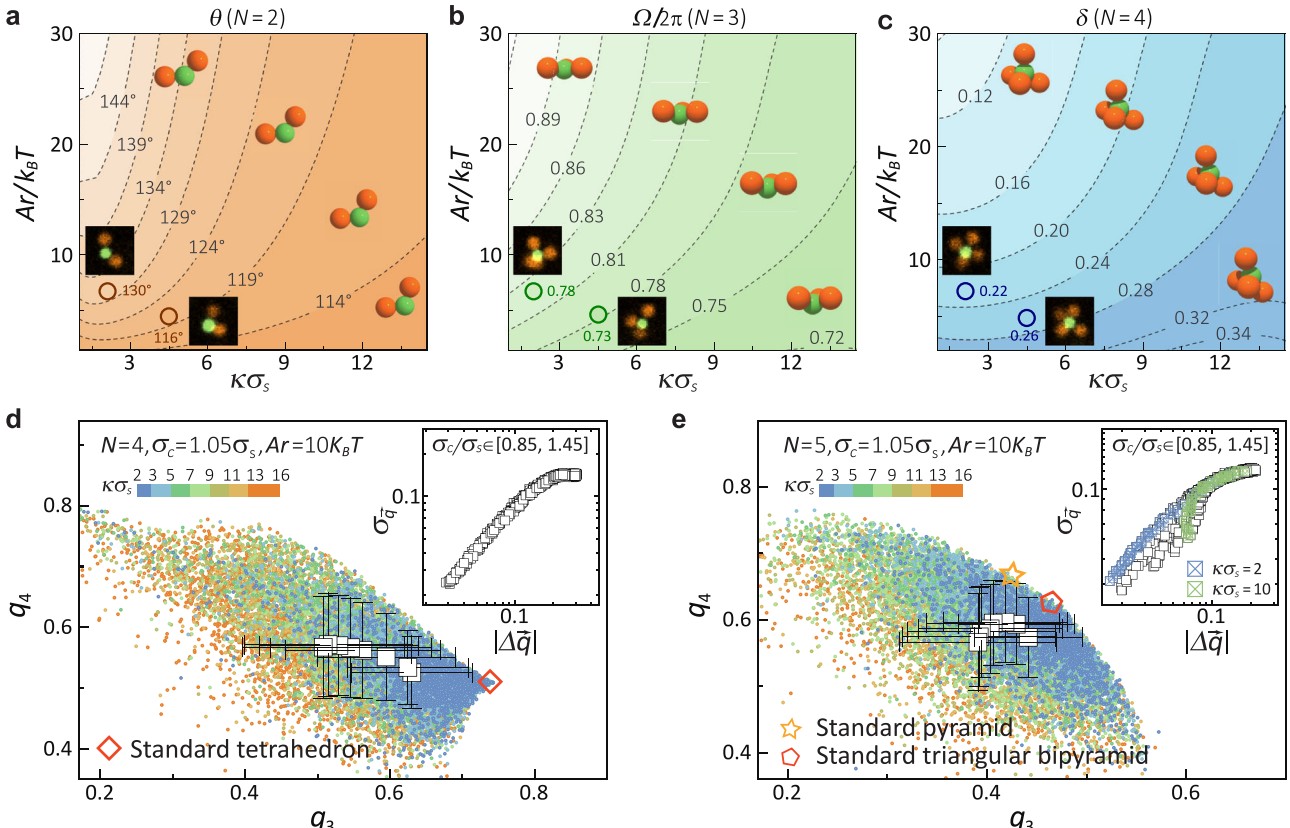

**Fig. 2 | Structure asymmetry and angular orders of N-CMs as a function of $\kappa\sigma_s$ and $\sigma_c$. a** Illustration of the asymmetric structure of 2-CMs through the bond angle $\theta$. **b** Illustration of the asymmetric structure of 3-CMs using the solid angle $\Omega$. **c** Illustration of the asymmetric structure of 4-CMs represented by $\delta$ (the deviation between the average and ideal configurations). Circular symbols represent the average configurations of over 50 independent CMs ($\sigma_c \approx \sigma_s$), obtained from experiments conducted under two different TBAB concentrations: 1 μM (left) and 5 μM (right). The confocal snapshots display representative configurations that closely resemble these average. Standard deviations of $\theta$ (2-CMs), $\Omega$ (3-CMs), and

the magnitude of configurational fluctuation around the average configuration $\Delta$ (4- and 6-CMs) are presented in Figs.S1d and e. **d** and **e**, Angular order development for 4- and 5-CMs depicted by $q_3$ and $q_4$. Square symbols with error bars represent the average and standard deviation (for $\sigma_c = 0.95\ \sigma_s$ and $A_r = 10\ k_BT$). Subpanels illustrate the $|\Delta\vec{q}|$-$\sigma_{\vec{q}}$ master curve for varying $A_r$ (5 to 75 $k_BT$), $\sigma_c$ (0.95 to 1.45 $\sigma_s$, incremented by 0.1 $\sigma_s$), and $\kappa\sigma_s$ (2 to 30). Note the bifurcation of 5-CMs labelled in blue symbols ($\kappa\sigma_s = 2$, towards triangular bipyramid) and green symbols ($\kappa\sigma_s = 10$, towards pyramid) in the inset of **e**. The standard triangular bipyramid is used as the reference configuration when calculating $|\Delta\vec{q}|$ for 5-CMs.

example, the asymmetry in 2-CMs. The potential energy minimum occurs when one particle is positioned with a relative angle to the other of $\theta = 180°$ for two freely moving and repulsive particles constrained on a spherical surface. However, the entropically favoured relative angle is $\theta = 90°$. Consequently, the dynamic structures inherently exhibit asymmetry. This simple yet universal principle, previously undisclosed, provides guidance for the design of asymmetric clusters at micro and nano scales.

We further quantified CMs' angular orders using the Steinhardt bond orientational order parameters, $q_l$ (see Methods for definitions). The configuration of 4-CMs gradually converges to a standard tetrahedron with a decrease in $\kappa\sigma_s$ (at $\sigma_c/\sigma_s = 0.95$), as illustrated by the $q_3$-$q_4$ distributions in Fig. 2d. To characterise the convergence, we defined a vector $\vec{q} = (q_3, q_4)$, with $q_3$ and $q_4$ as its two components, and calculated the deviation from the ideal configuration ($|\Delta\vec{q}|$) and the standard deviation of $\vec{q}$ ($\sigma_{\vec{q}}$). Interestingly, after a stage where the configurations are "liquid-like" (plateau of $\sigma_{\vec{q}}$), we observed a rapid, continuous improvement of angular orders ($|\Delta\vec{q}| \rightarrow 0$), as depicted by the $|\Delta\vec{q}|$-$\sigma_{\vec{q}}$ master curve in the inset of Fig. 2d (for $\sigma_c/\sigma_s \in [0.85, 1.45]$).

The rapid and continuous enhancement of angular order is marked by a swift convergence towards the ideal configuration. Notably, we emphasise the resemblance of these configurations to

those in hydrated ions that have recently been acknowledged for their dynamic structures and fluctuations around perfect angular symmetries[38]. This similarity hints at the potential for real-time exploration of complex ion-specific assembly behaviours. Similar behaviour is also evident for 3-CMs and 6-CMs (see Fig. S2a-b).

5-CMs exhibit intriguing polymorphic behaviour, distinguishing them from other CMs, as illustrated by the mixing of $q_3$-$q_4$ distributions in Fig. 2e. Close to the "liquid-like stage", the average configuration resembles an irregular pyramid with significant distortions. Angular order development towards the standard pyramid becomes evident at large $\kappa\sigma_s$, e.g., $\kappa\sigma_s = 10$ (see Fig. S3a), while ordering towards the triangular pyramid is favoured at low $\kappa\sigma_s$, e.g., $\kappa\sigma_s = 5$ (see Fig. S3b). The bifurcation pathways are more clearly visible in the $|\Delta\vec{q}|$-$\sigma_{\vec{q}}$ master curve shown in the inset of Fig. 2e. Note that we use standard triangular bipyramid as the reference configuration when calculating $|\Delta\vec{q}|$ for 5-CMs. A systematic calculation of the free-energy difference, $\Delta F$, defined as $(F_T - F_P)/k_BT$, between the two polymorphs supports the observation that the polymorphic feature is present over a wide range of $\sigma_c$ when $\Delta F \sim 0$, as depicted in Fig. S3c.

## The kinetics and pathway of CMs' formation
The formation kinetics of CMs in our system involves multiple and sequential reactions among the core droplet, satellite particles, and free particles, progressing from 1-CM to N-CMs, as schematically

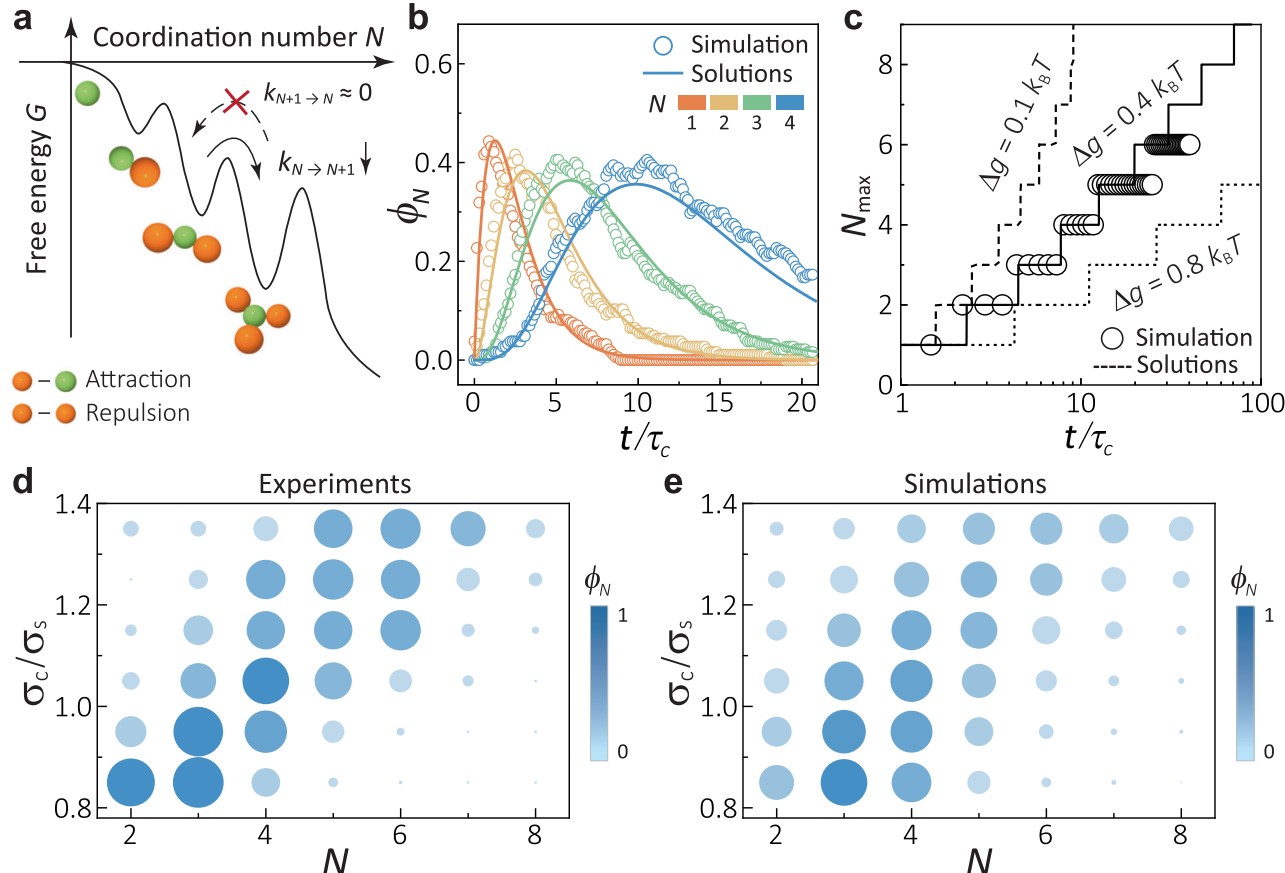

**Fig. 3 | The kinetics of *N*-CMs' growth. a** Schematic illustrating the sequential reaction process with respect to *N*. The forward transition barrier increases linearly with *N*, while the backward rate is nearly zero. **b** Evolution of $\phi_N(t)$ over time *t*. Molecular dynamic simulations (circles) align well with the solutions of $P_N(t)$. **c**, The most probable coordination number, $N_{max}$, versus time: dashed line ($\Delta g = 0.1 k_B T$), solid line ($\Delta g = 0.4 k_B T$), and dotted line ($\Delta g = 0.8 k_B T$). Optimisation of *N*-CMs production within a period is achievable with an appropriate $\Delta g$. **d, e** Distribution of *N* (indicated by the colour map and symbol sizes) with respect to $\sigma_c/\sigma_s$ in experiments (in **d**) and simulations (in **e**, 35 $\tau_c$).

illustrated in Fig. 3a. The free energy barrier associated with the addition of satellite particles increases due to particle-CM electrostatic repulsions. In contrast, detachment is irreversible owing to the strong core-satellite attraction. We defined the probability of forming *N*-CMs at time *t* as $P_N(t)$. The kinetic equation can be written as[20]:

$$\frac{dP_N(t)}{dt} = k_{N-1}P_{N-1}(t) - k_N P_N(t) \qquad (1)$$

Here, $k_N$ represents the transition rate for an *N*-CM to absorb another satellite: $k_N = k_0 \exp\left(-\frac{\Delta G_N}{k_B T}\right)$, where $k_0$ is the effective collision frequency between the core and free particles, and $\Delta G_N$ is the free energy barrier for a free particle to attach to a *N*-CM. Initially, $P_N(0) = 1$ for $N = 0$. We assume $\Delta G_N = N\Delta g$, where $\Delta g$ is a coefficient depending on $\sigma_c$, $\kappa\sigma_s$ and $A_r$. $P_N(t)$ can then be solved iteratively:

$$P_N(t) = \begin{cases} \exp(-k_N t), & N = 0 \\ \sum_{i=0}^{N} \frac{\prod_{m=0}^{N-1}(-k_m)}{\prod_{n=0, n\neq i}^{N}(k_i - k_n)} \exp(-k_i t), & N > 0 \end{cases} \qquad (2)$$

We also conducted molecular dynamic simulations using HOOMD[39]. For each core size, we run 200 replicas to investigate the temporal evolution of the fraction of *N*-CMs within the entire CM population, denoted as $\phi_N(t)$ (see Methods). The solutions for typical $N < 5$ in $P_N(t)$ ($\Delta g = 0.4 k_B T$) demonstrate good agreement with simulations, as illustrated in Fig. 3b. Note that $\kappa\sigma_s$ and $A_r$ approximately follow the contour

lines of $\delta$ in Fig. 2c for constant $\Delta g$. Deviation occurs for $N \geq 5$ when the assumption breaks down due to the core's accommodation capacity limit. The solutions suggest that optimising the production of *N*-CMs within a fixed period requires a specific $\Delta g$ value. For example, setting $\Delta g = 0.4 k_B T$ optimizes 6-CMs within 35 $\tau_c$ ($\tau_c \equiv 1/k_0$), as depicted in Fig. 3c. At various $\sigma_c/\sigma_s$, the experimentally observed distributions of *N*-CMs in Fig. 3d approximately align with simulations in Fig. 3e. The $\phi_N$ maxima in experiments are better than in simulations due to the much longer waiting time in experiments. Changing $\sigma_c$ can also perform the optimisations, consistent with a previous study[20,40].

## A kinetic tool box for designing CMs

Efficient production optimisation of *N*-CMs requires appropriate values of $\sigma_c$ and $\kappa\sigma_s$. However, we have also shown the need for sufficiently small $\sigma_c$ or $\kappa\sigma_s$ to stabilise regular CMs. Thus, efficiently producing regular CMs with maximum yield poses a challenge in pathway design. Inspired by our previous experimental study where colloidal particles crystallize within oil-in-water emulsions[34], we observed a decrease in salt concentration within the oil droplets due to ion diffusion into the surrounding water bath. This decrease is substantial in the first few hours after sample preparation, gradually slowing over time. This observation prompted us to explore the possibility that in-situ control over ionic strength could be a means modulate CM assembly kinetics.

Here, we propose a strategy and offer potential design examples. Our strategy involves a ramping protocol, which reserves $\kappa\sigma_s$ at a relatively high value with a waiting time optimizing irregular $(N-1)$-

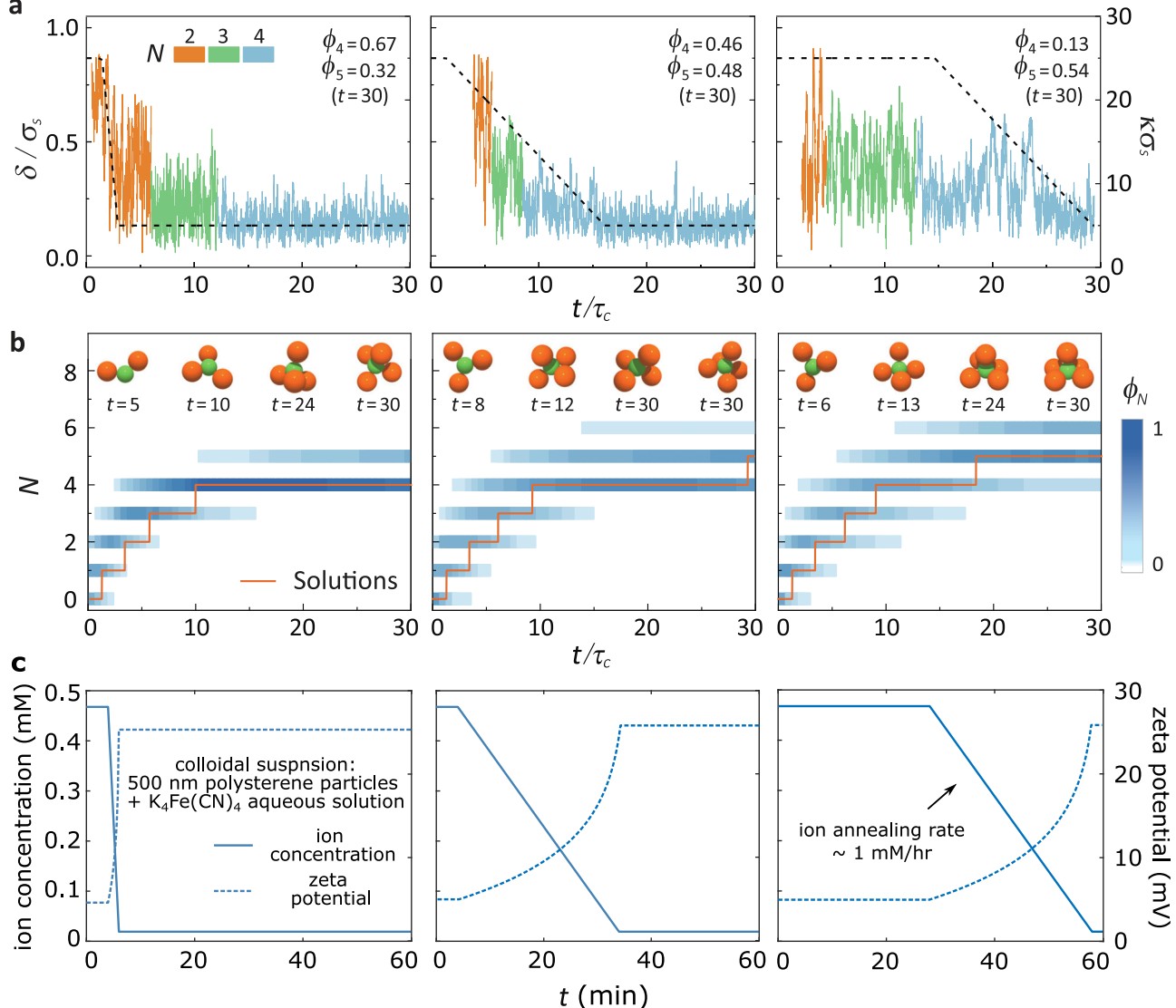

**Fig. 4 | The dependence of N-CMs' growth kinetics on ramping protocols derived from simulations. a** Three ramping protocols $\kappa\sigma_s(t)$ (black dashed lines) and the corresponding configurational fluctuations $\delta(t)/\sigma_s$ of $N$-CMs (coloured solid lines) with respect to the scaled elapsed time $t/\tau_c$. **b** Probability distributions of $N$-CMs corresponding to the three different ramping protocols in **a**. We set $A_r = 30 \, k_B T$ in molecular dynamic simulations. Solid lines represent the most probable $N$ from solutions of $P_N(t)$ ($\Delta g = 0.4 \, k_B T$). **c**, Tuning the ionic concentration (blue solid lines) and zeta potential (blue dashed lines) over time in a polystyrene colloidal suspension. Each subpanel corresponds to the respective ramping protocol illustrated in **a**.

CMs (e.g., 10 - 30 $\tau_c$ at $\kappa\sigma_s = 25$ for 4-CMs). Subsequently, $\kappa\sigma_s$ gradually lowers to a low value (e.g., $\kappa\sigma_s = 5$), ensuring that $N$-CM possess high angular order and the transition rate from 4-CMs to 5-CMs is not excessively low.

The waiting time and ramping rate are crucial parameters influencing the distribution of $N$-CMs. In Fig. 4a, b, we present simulation results for three different ramping protocols, analysing the corresponding deviation $\delta$ and $N$ over time. In the case of a fast ramping rate and short waiting time (left panels), $N_{max}$ remains at 4 for over 20 $\tau_c$ with a yield over 50%. For the same starting point but a lower ramping rate (middle panels), the growth becomes faster, resulting in almost equal yields of 4- and 5-CMs at the end of the simulation. Finally, with a proper waiting time and a slower ramping rate (right panels), the growth is even faster, and 5-CMs dominate at the end of the simulation. Solutions of $P_N(t)$ also accurately capture these trends and predict the most probable $N$ over time.

These comparisons indicate that the ramping should initiate when $(N − M)$-CMs are prevalent (M could be 1 or 2), even though their configurations deviate significantly from the standard configuration.

Otherwise, the free-energy barrier for another satellite to associate with would be too significant. Following this, an appropriate quench rate towards conditions favouring symmetric $N$-CMs should be applied to condense the distribution to $N$ with the maximum yield. The precise quantitative relationship between the distribution of $N$, ramping rates, and waiting time remains further exploration. Our theoretical framework can serve as a valuable tool for guiding the design of such ramping protocols.

To provide a more concrete instance of how the physical parameters in the experimental system should be tuned to align with the proposed ramping protocol, we present an example in Fig. 4c. The system under consideration is a 500 nm polystyrene colloidal suspension in a $K_4Fe(CN)_4$ aqueous solution, in which the relationship between zeta potential and ion concentration has been previously measured[41]. In a dilute system, the collision time $\tau_c$ can be estimated as $\tau_c \sim l_{free}^2/D$, where $l_{free}$ is the mean free path and $D$ is the diffusion constant. At a 0.1% volume fraction, the estimated collision time is around 2 minutes, which is significantly longer than the time required for ion equilibration. By mapping the screening length $\kappa$ and ion

concentration through the equation $\kappa^{-1} = \sqrt{4\pi\lambda_b n_{ion}}$, where $\lambda_b$ is the Bjerrum length and $n_{ion}$ is the effective ion number density, we convert the ramping protocol in Fig. 4a to the corresponding parameters shown in Fig. 4c. In this annealing protocol, the ion concentration decreases by an order of magnitude, from 0.5 mM to 0.01 mM, over a period of several minutes to hours, resulting in an annealing rate of approximately 1 mM/hr. It is important to note that in practice, the required ramping rate may need to be even slower, since the zeta potential increases from 5 mV to 25 mV as the salt concentration decreases. This increase in zeta potential should raise the energy barrier for satellite attachment and thereby slow the assembly kinetics. This effect should be considered in Eq. (1) for a more accurate solution.

Additionally, to implement this concept experimentally, we propose a two-step approach to control the quench rate of the ionic strengths through the size of oil-in-water emulsion droplets, as illustrated in the flow chart in Fig. S4. Alternatively, dynamic tuning of ionic strengths in experiments can be achieved by adjusting the solvent fraction through dilution or evaporation[42,43]. We hope this experimental scheme will inspire material engineers to apply the proposed theoretical strategy for controlling the in-situ self-assembly dynamics of CMs.

## Summary and Outlook

In this study, we systematically explored the dynamic structures and formation kinetics of CMs by combining 3D real-space imaging and computer simulations. We utilised a colloidal-emulsion model system designed to form micrometre-sized dynamic CMs with tunable electrostatic interactions. The key outcomes of our research unveil the intrinsic asymmetry in the dynamic structures of CMs, with their angular symmetry evolving through continuous and sequential ordering from a liquid-like configuration, contrasting with a sharp first-order transition. Notably, our findings also highlight the polymorphic nature of 5-CMs, introducing a richer diversity of behaviours compared to other CMs.

The emergence of asymmetric dynamic structures arises as a collective mode of motion involving a few or several satellite particles on a curved surface. This asymmetric nature in CMs' dynamic structures is considered universal for CMs whose angular order results from satellite-satellite repulsive interactions through a Thomson-like mechanism. This behaviour, influenced by thermal fluctuations, reflects a balance between energy and entropy. This asymmetry is more pronounced as the effective satellite-satellite repulsive interaction becomes shorter-ranged. In cases where the angular order is a result of a strong core-satellites anisotropic interaction, such as patchy interactions, the motion of satellite particles becomes localised rather than collective, and the asymmetric nature is anticipated to be less pronounced.

These findings offer valuable insights and potential strategies for researchers engaged in assembling hierarchical structures using flexible colloidal molecules. It is worth noticing that configurational fluctuations play a crucial role in shaping self-assembly dynamics, potentially leading to multi-step solid-solid transitions and bifurcation of the growth[44–49]. Nevertheless, utilising fluctuations to engineer a smoother assembly pathway toward the target structure remains a significant challenge. The insights into configurational distributions arising from thermal fluctuations offer valuable knowledge for designing pathways towards complex hierarchical structures.

On the kinetic side, achieving a CM with a specified angular order is challenging due to the multiple and sequential reaction process among colloidal particles. To overcome this difficulty, we proposed a method that dynamically adjusts the screening of charge-charge interaction, guiding the kinetics of CM ordering toward the target structure. Our kinetic model is versatile and can be applied to systems with similar interaction frameworks. By mapping transition rates to experimental parameters, the model facilitates the identification of the

parameter space that yields the highest yield in the minimum time. Furthermore, we highlight that there has been growing interest in utilising non-equilibrium pathways to obtain "shortcuts" towards desired states[50–52]. Our colloidal molecule system provides a complex multi-dimensional parameter space to validate these theoretical models from both computational and experimental perspectives.

## Methods

### Experiments

Our CMs are composed of satellite PMMA colloidal particles with a diameter of approximately $\sigma_s \approx 1.2\,\mu m$ and a central water(20% glycerol)-in-oil emulsion droplet with a diameter $\sigma_c$ ranging from 0.8 μm to 1.6 μm. The formation of these CMs involves mixing PMMA colloidal suspensions with emulsions.

The colloidal suspension comprises polymethyl methacrylate (PMMA) particles and grafted with poly(12-hydroxystearic acid). These particles are dispersed in a density and refractive-index matching oil solvent, a mixture of 70% cyclohexyl bromide (CHB) and 30% cis-decalin (volume ratio), at a 5% volume fraction. The measured particle diameter is $\sigma_s \approx 1.2\,\mu m$ with a size polydispersity of 2.5%. The PMMA particles exhibit a negative charge in the solvent, producing a soft interparticle repulsion. The experimentally determined pair potential, $U_{pp}(r)$, is estimated using the equation:

$$U_{pp}(r) = \frac{Q^2}{\sigma_s(1+\frac{\kappa\sigma_s}{2})^2 \epsilon_s} \exp[-\kappa\sigma_s(r/\sigma_s - 1)]/(r/\sigma_s),\ r > \sigma_s \quad (3)$$

Here, $Q$ represents the surface charge of the colloidal particle, and $\epsilon_s$ is the dielectric constant of the solvent. The term $\kappa^{-1}$ denotes the screening length, and $\frac{Q^2}{\sigma_s(1+\frac{\kappa\sigma_s}{2})^2 \epsilon_s}$ is a surface potential parameter.

We prepared the water(20% glycerol)-in-oil emulsions following the same method outlined in Ref. 33. Tetrabutylammonium bromide (TBAB) salt was introduced to modify the Debye length within the oil mixture (0 - 5 μM). The water phase in the emulsion consisted of a 20% glycerol aqueous solution (in volume), and 10 mM sodium dodecyl sulfate (SDS) was added to prevent particle adhesion onto the water-oil interface. The volume ratio between glycerol and the CHB-decalin solution was maintained at 1:40. By subjecting the sample to 2 minutes of sonication, water droplets ranging in size from hundreds of nanometers to 5 micrometres were achieved. Larger droplets were eliminated through coalescence using a centrifuge (100 $g$, where $g$ is the gravitational acceleration). It is important to note that the water-glycerol droplets also carry a negative charge in the solvent.

The emulsions were subsequently combined with PMMA colloidal suspensions using a vortex mixer with a stoichiometry ratio of approximately 200:1 between particles and emulsion droplets. The particle-droplet interaction is characterised by a long-range, charge-charge repulsion. Moreover, attractive forces, specifically the charge-image-charge interaction and hydrophobic one, dominate the particle-droplet interaction at short-range[33]. The resulting mixture was left at room temperature for 24 hours to ensure the formation of a stable suspension of colloidal molecules.

We used a Leica SP8 fast confocal microscope equipped with a 63 × /1.40-NA oil-immersion objective to observe the CMs. PMMA particles, dyed with Nile red, were excited at 552 nm, and their emission was detected between 580 and 700 nm. Water droplets, dyed with fluorescein sodium salt, were excited at 488 nm, with emission detected between 500 and 540 nm. The scanning speed was 28 frames per second, with each frame consisting of 128 pixels × 128 pixels (voxel size: 134 nm/pixel). The spacing between consecutive frames in the $z$ direction was 250 nm. Since the height of a typical CM is less than 4 μm, fewer than 16 images (taking around 576 milliseconds) were required to scan through the entire structure, enabling us to capture the instantaneous configuration. Given the wide range of droplet sizes

in our system, we employed an advanced particle-tracking method to identify these droplets. This algorithm, detailed in Refs. 53,54, is based on the Scale Invariant Feature Transform (SIFT), enabling precise localization of particles from three-dimensional (3D) confocal images without making any assumptions about the target sizes.

The values of $Q$ and $\kappa^{-1}$ were estimated from the conductivity $\sigma$ and zeta potential $\zeta$, following the measurement protocol outlined in Ref. 37. The measured values for $(\sigma,\zeta)$ were (0.2 nS/cm, 14 mV) at 1 µM TBAB and (1.0 nS/cm, 12 mV) at 5 µM TBAB. Using the following relationships: $\kappa^{-1} = \sqrt{\epsilon_r\epsilon_0 k_B T / 6\pi\eta a\sigma}$, $Q = 2\pi\epsilon_r\epsilon_0\sigma_s(1 + \kappa\sigma_s/2)\zeta$, where $\eta$ is the solvent viscosity (2 mPa · s), $a$ is the hydrodynamic radius of the solute molecule (0.5 nm), and $\epsilon_r$ is the relative dielectric constant ($\epsilon_r \approx 4$), we estimated $(\kappa^{-1}, Q,)$ to be (630 nm, 45 e) at 1 µM TBAB and (270 nm, 63 e) for 5 µM TBAB. The corresponding values of $(\kappa\sigma_s, A_r)$ were approximately (2.0, 6.5 $k_B T$) at 1 µM TBAB and (4.5, 4.7 $k_B T$) at 5 µM TBAB.

## Simulations

We employed canonical Monte Carlo simulations to sample the equilibrium distribution of CMs' configurations. Satellite particles were confined to a spherical surface with a diameter denoted as $\bar{\sigma} \equiv (\sigma_s + \sigma_c)/2$, where $\sigma_s$ and $\sigma_c$ represent the diameter of satellite particles and core droplets, respectively.

The influence of gravity on our system was found to be negligible. The density of the CHB-decalin solvent (1.18 g/cm³) was closely matched to the average density of CMs. The density difference between the PMMA particles and the solvent was less than 1%, while the difference between the PMMA particles and the water droplets was approximately 3%. The gravitational length $h$ was estimated using the relation: $\frac{\pi}{6}\Delta\rho g\sigma^3 h \sim k_B T$, which yields $h \sim 20$ µm, significantly larger than the particle size of 1.2 µm. Moreover, as shown in our SI videos, the CMs rotate freely without any observable accumulation of PMMA particles at the bottom. We also performed a centrifugation experiment at 1000 $g$ for 5 minutes with a density-matched sample, where no noticeable sedimentation or density gradient was observed along the $Z$ direction.

The particle-particle pair potential in simulations, $U_{pp}(r)$, is similar to that in experiments:

$$U_{pp}(r) = A_r \exp[-\kappa\sigma_s(r/\sigma_s - 1)]/(r/\sigma_s), \quad r > \sigma_s, \tag{4}$$

where the adjustable parameter $A_r$ represents the surface potential parameter of the satellite particle, and $\kappa$ corresponds to the inverse of the Debye screening length.

To study the dynamics of growing colloidal clusters, we used the HOOMD package for molecular dynamics simulations. We used the same $U_{pp}$ in experiments and MC simulations. The interaction $U_{pd}$ between a satellite particle and a core droplet comprises a weak, long-range repulsive component and a much stronger attractive component. To speed up the simulation, we neglected the long-range repulsive part. In simulations, we use the following simple, attractive pair potential for $U_{pd}(r)$:

$$U(r) = \frac{A_a}{r^6}, \quad r > \bar{\sigma} \tag{5}$$

The parameter $A_a$ is negative, and its magnitude is large enough to prevent the satellite-core detachment. We initiated each simulation with one core droplet and 50 satellite particles in the box. The volume fraction of particles was set to 8 %, an order of magnitude higher than that in experiments. The diameter of satellite particles is 1 µm, while the core size ranges from 0.85 µm to 1.45 µm with an increment of 0.10 µm. The dynamic viscosity was set to be equal to that of water, which is 1.0 mPa · s. For each core size, we ran 200 replicas to study how the fraction of $N$-CMs in total CMs, denoted as $\phi_N$, evolves with time. Each simulation replica evolved for a duration of 50 seconds.

## Configurations and angular orders of CMs

To determine the maximum superimposition of experimental configurations, we implemented a progressive alignment algorithm. Initially, all CMs were normalised and translated to a common position, aligning their droplet coordinates with the origin. Subsequently, a CM was randomly selected as the initial configuration. To achieve the best alignment, we employed the Kabsch algorithm, which computes an optimal rotation matrix ($R$) to minimise the root mean squared deviation (RMSD) between the initial and subsequent configurations[35]. RMSD is given by the following formula:

$$\text{RMSD} = \sqrt{\frac{\sum_i |\mathbf{r}_i^0 - \mathbf{r}_i^1|^2}{N}}, \tag{6}$$

where $\mathbf{r}_i^0$ and $\mathbf{r}_i^1$ are the satellite particle positions of the initial and subsequent configurations, respectively.

The average configuration was determined by iteratively incorporating experimental configurations of $N$-CMs into the entire population until convergence was achieved. Subsequently, each new configuration was compared to the previous population average and adjusted to its optimal orientation through rotation. The average configuration was then updated by integrating the contribution of the new configuration. This process was repeated multiple times on our set of experimental configurations until the RMSD values reached convergence. The final configuration distributions were derived from the coordinates obtained during the last iteration.

It is crucial to note that the Kabsch algorithm is specifically applicable to sets of paired points. To establish pairs between a new CM and the iterative configuration, RMSD values were calculated for all possible permutations of its satellites with the iterative configuration. Subsequently, we identified the positions that resulted in the minimal RMSD value.

To quantitatively assess the impact of various simulation parameters on the configuration of $N$-CMs, we employed two parameters: the deviation between the average and standard configurations, denoted as $\delta$, and the magnitude of fluctuations around the average configuration, denoted as $\Delta$. The parameter $\delta$ is defined as follows:

$$\delta = \sqrt{\frac{\sum_i |\bar{\mathbf{r}}_i^e - \mathbf{r}_i^0|^2}{N}}. \tag{7}$$

Here, $\bar{\mathbf{r}}_i^e$ represents the mean position of the $i$-th satellite of $N$-CM, while $\mathbf{r}_i^0$ denotes the position of the satellite of the standard configuration. $\Delta$ is defined as

$$\Delta = \sqrt{\frac{\sum_i \sum_j |\mathbf{r}_{ij}^e - \bar{\mathbf{r}}_i^e|^2}{MN}}. \tag{8}$$

Here, $\mathbf{r}_{ij}^e$ represents the position of the $i$-th satellite of the $j$th $N$-CMs observed, and $M$ is the total number of $N$-CMs observed. Figs. S1d and e show the evolution of $\Delta$ and $\delta$ with respect to $\kappa\sigma_s$ for 6-CMs, similar to the trend observed in 4-CMs.

To evaluate the angular symmetry of CMs, we computed the bond orientation order parameters $q_3$ and $q_4$ of CMs and studied their distributions. The bond orientation order parameter $q_l$ is defined as

$$q_l = \sqrt{\frac{4\pi}{2l+1}\sum_{m=-l}^{l}|q_{lm}|^2}, \tag{9}$$

$$q_{lm} = \frac{1}{N}\sum Y_l^m(\theta,\phi), \tag{10}$$

where $\theta$ and $\phi$ are the azimuthal and polar angles of satellite particles concerning the core, respectively, and $Y_l^m$ are the spherical harmonics. Figs. S2a-c illustrate the evolution of the bond orientation order

parameters for 3-CMs and 6-CMs, mirroring the trend observed in 4-CMs.

To determine whether a 5-CM aligns more closely with a triangular bipyramid or a pyramid configuration, we rotate them to achieve maximal superimposition with standard triangular bipyramid and pyramid configurations having distinct heights. The minimal RMSD of pyramid configurations is then compared with the RMSD of a standard triangular bipyramid. A smaller RMSD indicates proximity to the correspondent category.

Figs. S3a and b present the details of the two bifurcation pathways. At $\kappa\sigma_s = 10$, the average configuration of 5-CMs develops towards the pyramid as the core diameter $\sigma_c$ decreases to 0.85 (Fig. S3a). However, when $\kappa\sigma_s = 5$, the development of 5-CMs leans more towards the triangular bipyramid as $\sigma_c$ reduces (Fig. S3b). This bifurcation stems from the free-energy difference between the two polymorphs of 5-CMs at different $\kappa\sigma_s$ and $\sigma_c$. Due to such a bifurcation, achieving a pure triangular bipyramid for 5-CMs through a pathway of reducing $\kappa\sigma_s$ becomes challenging. A more feasible pathway involves reducing $\sigma_c$ at $\kappa\sigma_s < 5$.

## Comparison between experiments and simulations

To match the experimental conditions with the simulation parameters, we explored a parameter space defined by two fitting parameters: $A_r$ and $\kappa\sigma_s$. The parameter $A_r$ was varied from $5\,k_BT$ to $30\,k_BT$ in increments of $5\,k_BT$, while $\kappa\sigma_s$ ranged from 2 to 16 in steps of 2. To quantify the similarity between experimental and simulated configurations, we defined a parameter $\delta^0$ that accounts for both the average and the spread of the configurations. This parameter is defined as:

$$\delta^0 = \sqrt{\delta_m^2 + \delta_s^2}, \tag{11}$$

where

$$\delta_m = \sqrt{\frac{\sum_i |\bar{\mathbf{r}}_i^e - \bar{\mathbf{r}}_i^s|^2}{N}}, \tag{12}$$

and

$$\delta_s = \sqrt{\frac{\sum_i (\Delta\mathbf{r}_i^e - \Delta\mathbf{r}_i^s)^2}{N}}. \tag{13}$$

Here, $\delta_m$ captures the difference in the mean positions of vertices between experiments and simulations, while $\delta_s$ describes the disparity in positional fluctuations. $\bar{\mathbf{r}}_i^e$ and $\bar{\mathbf{r}}_i^s$ represent the mean positions of the $i$-th vertex of $N$-CMs in experiments and simulations, respectively. Similarly, $\Delta\mathbf{r}_i^e$ and $\Delta\mathbf{r}_i^s$ denote the standard deviation of the $i$-th vertex's position relative to its mean in both experiments and simulations. Before calculating $\delta^0$, we aligned the configurational distributions from experiments and simulations by rotating them to achieve maximal overlap. For each salt concentration, we computed the mean $\delta^0$ over a wide range of core particle sizes $\sigma_c$ (from $0.85\,\sigma_s$ to $1.15\,\sigma_s$) and satellite numbers $N$ (ranging from 2 to 6) to identify the optimal simulation parameters. The optimal parameters were found to be $A_r = 7.5\,k_BT$ and $\kappa\sigma_s = 2$ for 1 μM TBAB, and $A_r = 5\,k_BT$ and $\kappa\sigma_s = 8$ for 5 μM TBAB. Under these conditions, $\delta^0$ was generally less than $0.1\,\sigma_s$. Although the precise values differ from those estimated from experimental measurements, the trend of asymmetry indicated by our fit aligns with experimental observations. We hypothesize that this deviation may result from the limitations of the Yukawa model at higher salt concentrations, where image charge effects become significant.

## Data availability

The raw data that support the findings of this study are available from the corresponding authors upon request. Source data of figures are provided with this paper as a Source Data file. Source data are provided with this paper.

## Code availability

The codes that are used to generate results in the paper are available from the open repository: https://doi.org/10.6084/m9.figshare.28489886.v1.

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

## Acknowledgements

This work was supported by the National Key R&D Program of China (2022YFA1503501, W. Li), the National Natural Science Foundation of China (Nos 12425503, 12174071 and 12035004, P. Tan), the Space Application System ofChina Manned Space Program (KJZ-YY-NLT0501, P. Tan), the Innovation Program of Shanghai Municipal Education Commission (2023ZKZD06, P. Tan), and Shanghai Pilot Program for Basic Research-FuDan University (22TQ003, P. Tan; 21TQ008, W. Li). H. Tanaka acknowledges Grant-in-Aid for Specially Promoted Research (JP20H05619) from the Japan Society of the Promotion of Science (JSPS).

## Author contributions

P.T., W.L., H. Tanaka and Z.N. conceived and supervised the research and made data analysis, modelling, and wrote the paper, H.F., Q.G., Y.R. and Y.C. performed experiments, simulations and data analysis, H.Tong and J.H. made data analysis.

## Competing interests

The authors declare no competing interests.
