## [Transparent Peer Review file · Nature Communications]

Dynamic and asymmetric colloidal molecules

Corresponding Author: Professor Peng Tan

Version 0:

Reviewer comments:

Reviewer #2

(Remarks to the Author)

The manuscript of Fang et al presents a detailed study of the dynamic structures and kinetics of assembly of flexible colloidal molecules via confocal microscopy experiments and simulations. The authors present detailed experiments on 3D imaging and tracking of different numbers of satellite PMMA particles surrounding a central water-glycerol droplet in a refractive index matched Cyclohexyl bromide-cis-decalin solvent. These particles are investigated at different salt (TBAB) concentrations which controls the Debye length and thus the electrostatic interactions. The experiments are complimented by extensive simulations of the dynamics of the same clusters. This shows that the arrangement of the satellite particles is asymmetric as with the increased screening showing the solutions to the Thompson model no longer hold. In addition, the authors investigate the kinetics of the colloidal molecule growth and finally, the authors show how the growth kinetics can be controlled via ramping the salt concentration/screening length. The most interesting result is that by understanding the role of screening and kinetics on the cluster configurations, the authors propose an assembly pathway that can control the colloidal molecule configuration outcome, despite ending at the same conditions. Overall, the manuscript is well written, and the findings highlight the importance of size ratio and screening length on the final assembly pathway of colloidal molecules and opens the possibility to “design” assembly pathways of hierarchical structures. I expect the results to be of interest to others in the field but also to a larger community working on self-assembly of complex materials ranging from chemists to biologists and physicists. Therefore, I recommend publication of the manuscript in Nature Communications.

However, before publication I would recommend the authors to address the following issues:

(1) The main control discussed in the manuscript is that via screening of the electrostatic interactions between the satellite particles. However, in the introduction and the main text about the experiments the control over the salt concentration and the corresponding Debye length is not very clear (I had to read it 3 times). As the authors claim that “By mapping transition rates to experimental parameters, the model facilitates the identification of the parameter space that yields the highest yield in the minimum time.”. They indicate that final presented simulation model can provide the required salt concentration range for experiments to allow controlled production of asymmetric colloidal molecules, I would recommend the authors to make it much clearer which salt concentrations and Debye lengths and surface charge need to be achieved. The authors should also provide some values for the Debye length and Q in the experiments and explained how they determined this – i.e. conductivity measurements or estimations based on TBAB concentration. They should provide the exact concentration and values used.

For instance, at page 4:

“Tetrabutylammonium bromide (TBAB) salt is introduced into the solvent to modify κ^{-1} and Q .” include κ^{-1} and Q range
“At 5 μM TBAB,” include κ^{-1}

(2) The authors did very impressive experiments on 3D tracking of real experimental clusters as presented in Fig 1. It is clear that the angle and asymmetry can be measured for the clusters (Fig. 1d). It would be great if some of the experimental results could be compared with the extensive simulation results on the asymmetry in Fig 2. This would make it clearer to the reader how much of the simulations are verified by experiments.

(3) The CLSM imaging description is in the current form not reproducible. The authors should provide information about the fluorescent dyes used (Nile – is not sufficient information). The laser excitation wavelengths and detection range, also the xy-scan speed and voxel size (or image size and volume) should be provided. It is also unclear how long the clusters were observed for the data presented in (Fig 1d).

(4) In Fig 3c the authors present the dominant colloidal molecule N as a function of time for different Δg . The authors state that “For example, setting $\Delta g = 0.4 \text{ kBT}$ optimizes 4-CMs within $35 \tau_c$ ($\tau_c \equiv 1/k_0$), as depicted in Fig. 3c.” However, my understanding of the data presented in Fig 3c is that 6-CMs are achieved at $t = 35 \tau_c$. Thus this claim seems to be incorrect, or the authors should explain how they came to this conclusions more clearly.

Other small points the authors should correct:

- The authors seem to have made a very unfortunate auto-correct, they talk about cis-deCLINE in the main manuscript, this should be cis-deCALIN. The authors should correct this. (In methods it is correct).

Reviewer #4

(Remarks to the Author)

I have read the manuscript with interest and it contains sound, interesting and original work. However, regarding, the question whether this work is suitable for publication in NC I recommend to not publish. The main reason is that the theoretical idea of changing the ionic strength during the assembly process well defined angles can be obtained, unfortunately lacks experimental validation.

• What are the noteworthy results?

The authors have shown interesting measurements and simulations with regards to the formation of assemblies of colloidal particles attached to an emulsion droplet. This type of process is relevant to create colloidal molecules. A challenge remains for these structures to get well defined angles between the adsorbed colloids. This paper aims to shed light on the root cause and possible resolution of this challenge. In my view the most relevant information is the measurement of the angular information of the colloids with respect to the emulsion droplet center, and the asymmetry that these angles have.

• Will the work be of significance to the field and related fields? How does it compare to the established literature? If the work is not original, please provide relevant references.

I accepted to review this paper based on the mentioning in the abstract of gradually changing the salt concentration during assembly: "We further develop a novel method to guide the ordering of CMs towards a desired structure by dynamically adjusting the ionic strength in the solvent during the ordering process." This is a nice idea but theoretically reasonably straightforward. Unfortunately the paper does not contain experiments using gradual change in salt concentration only simulation, because of this I felt a little misguided by the abstract. I recommend stating more clearly in the abstract that at this point it is based on simulations not experiment.

Although the topic does contribute to a degree to the understanding of mechanisms leading to the formation of colloidal molecules having well controlled bond angles this topic is been covered in the past with more success in practical terms as such molecules have been synthesized as seen in: J.D. Kraft, J. Groenewold and W.K.Kegel, Colloidal molecules with well-controlled bond angles, *Soft Matter* 5(20), pages 3823-3826 2009

• Does the work support the conclusions and claims, or is additional evidence needed?

There is no mention of the role of gravity on the dynamics of the particles. In my view this should be addressed, as this may bias the angles. If deemed important it should be included in the model. If not important an argument or measurement should be given to show the irrelevance of gravity.

As far as the modelling is concerned there is definitely an effort to model the dynamics. For the adsorption / desorption kinetics a nice validation is shown in Fig 3d and e. In Fig. 2 shows (as far as I understand) only simulations. In principle there are measurements to compare them with. Is there a specific reason why this is not shown?

• Are there any flaws in the data analysis, interpretation and conclusions? Do these prohibit publication or require revision?
No flaws.

• Is the methodology sound? Does the work meet the expected standards in your field?

Sound methodology

• Is there enough detail provided in the methods for the work to be reproduced?

Yes

Jan Groenewold, Utrecht University

Version 1:

Reviewer comments:

Reviewer #2

(Remarks to the Author)

While the authors have addressed all my concerns adequately and improved the manuscript, the revised manuscript also contains changes in response to reviewer 4 which did not improve the manuscript, and I do not recommend publication in the current form.

I would like the authors to address the following issue:

Reviewer 4 has pointed out correctly that an experimental validation of the suggested pathway obtained via simulations is missing. To address this point, the authors have now added a panel (c) to figure 4. Although, I appreciate the effort of the detailed suggested preparation steps for their system, I find it does not improve the manuscript. The panel contains a lot of

text, is not easy to understand and contains an extremely specified pathway of a preparation protocol not presented earlier in the paper. Hence, I find it distracts from the main message of the manuscript and does not convey the message that their findings provide guidance for an experimental pathway that would allow one to obtain controlled colloidal molecules from charged particles by simple tuning of the salt concentration during the assembly process.

I would recommend the authors to remove this specific panel (maybe add it to the supplementary information) and instead provide a more general protocol for controlled colloidal molecule preparation by specifying which particle surface charge (zeta potential) and salt concentration change rate would be needed to be achieved under certain assembly kinetics in an experiment. This would be much more impactful for the field instead of a very complex pathway that can only benefit their specific system. Especially, as it is well known that there are many different particle systems and many more ways to achieve a change in salt concentration, as has been abundantly shown for colloids. Just as an example, in a very recent work on ionic colloidal crystals an in situ change in salt concentrations was used to follow the melting process, and this simply achieved via evaporation of the solvent (Zhang et al. Nature Materials volume 23, pages 1131–1137 (2024)).

Seeing that the authors have provided a detailed protocol for the experiment already, I would like to ask the authors to perform the proposed experiment. I would find their findings much more impactful (and very exciting) if it was experimentally verified that their proposed method works.

Other small comment:

For figure 2 the reference to panel d and e are missing

Reviewer #4

(Remarks to the Author)

I am satisfied with the changes made by the authors. I am somewhat doubtful about the significance of this work, as can be seen from my first review report, and the adaptations have not changed this. However all other points are satisfactorily addressed and I recommend to accept the paper without further adaptations.

Version 2:

Reviewer comments:

Reviewer #2

(Remarks to the Author)

The authors have adequately addressed my comments in the revised version. The more general route including the specific route for their system in the SI clarify how to such systems could be achieved experimentally. I recommend publication of the manuscript in Nature communications.

Replies to the comments of Reviewer #2:

The manuscript of Fang et al presents a detailed study of the dynamic structures and kinetics of assembly of flexible colloidal molecules via confocal microscopy experiments and simulations. The authors present detailed experiments on 3D imaging and tracking of different numbers of satellite PMMA particles surrounding a central water-glycerol droplet in a refractive index matched Cyclohexyl bromide-cis-decalin solvent. These particles are investigated at different salt (TBAB) concentrations which controls the Debye length and thus the electrostatic interactions. The experiments are complimented by extensive simulations of the dynamics of the same clusters. This shows that the arrangement of the satellite particles is asymmetric as with the increased screening showing the solutions to the Thompson model no longer hold. In addition, the authors investigate the kinetics of the colloidal molecule growth and finally, the authors show how the growth kinetics can be controlled via ramping the salt concentration/screening length. The most interesting result is that by understanding the role of screening and kinetics on the cluster configurations, the authors propose an assembly pathway that can control the colloidal molecule configuration outcome, despite ending at the same conditions. Overall, the manuscript is well written, and the findings highlight the importance of size ratio and screening length on the final assembly pathway of colloidal molecules and opens the possibility to “design” assembly pathways of hierarchical structures. I expect the results to be of interest to others in the field but also to a larger community working on self-assembly of complex materials ranging from chemists to biologists and physicists. Therefore, I recommend publication of the manuscript in Nature Communications.

However, before publication I would recommend the authors to address the following issues:

(1) The main control discussed in the manuscript is that via screening of the electrostatic interactions between the satellite particles. However, in the introduction and the main text about the experiments the control over the salt concentration and the corresponding Debye length is not very clear (I had to read it 3 times). As the authors claim that “By mapping transition rates to experimental parameters, the model facilitates the identification of the parameter space that yields the highest yield in the minimum time.” They indicate that final presented simulation model can provide the required salt concentration range for experiments to allow controlled production of asymmetric colloidal molecules, I would recommend the authors to make it much clearer which salt concentrations and Debye lengths and surface charge need to be achieved. The authors should also provide some values for the Debye length and Q in the experiments and explained how they determined this – i.e. conductivity measurements or estimations based on TBAB concentration. They should provide the exact concentration and values used. For instance, at page

4: "Tetrabutylammonium bromide (TBAB) salt is introduced into the solvent to modify κ^{-1} and Q ." include κ^{-1} and Q range "At 5 μM TBAB," include κ^{-1}

Response: We apologize for the lack of clarity in the original manuscript regarding the role of salt concentration in tuning electrostatic interactions. Additionally, we regret not explicitly providing the values for the Debye length (κ^{-1}) and surface charge (Q) used in our experiments.

The interaction between charged colloidal particles in our system is modeled by a hard-core screened Coulomb potential:

$$U_{PP}(r) = \frac{Q^2}{(1+\kappa\sigma_s/2)^2\epsilon_s r} \exp[-\kappa\sigma_s(r/\sigma_s - 1)].$$

Here, the salt concentration controls both the surface charge (Q) and the Debye length (κ^{-1}). As the salt concentration increases, both κ^{-1} and Q increase. This adjustment allows us to effectively control the range of electrostatic repulsion between the colloidal particles.

The values of κ^{-1} and Q were estimated using the conductivity σ and zeta potential ζ , measured following the protocol described in Ref. [1]. The measured values for (σ, ζ) were (0.2 nS/cm, 14 mV) at 1 μM TBAB and (1.0 nS/cm, 12 mV) at 5 μM TBAB. Using the relationships $\kappa^{-1} = \sqrt{\epsilon_0\epsilon_r kT/6\pi\eta a\sigma}$ and $Q = 2\pi\epsilon_0\epsilon_r\sigma_s(1 + \kappa\sigma_s/2)\zeta$, where η is the solvent viscosity, a is the hydrodynamic radius of the solute molecule, and ϵ_r is the relative dielectric constant. We estimated (κ^{-1}, Q) to be (630 nm, 45 e) at 1 μM TBAB and (270 nm, 63 e) for 5 μM TBAB. The corresponding values of $(\kappa\sigma_s, Ar)$ were approximately (1.9, 6.5 kT) at 1 μM TBAB and (4.5, 4.7 kT) at 5 μM TBAB.

From our simulations, we identified that effective control of the self-assembly of asymmetric CMs requires the range of $\kappa\sigma_s$ to ideally fall between 1 and 20, while the surface potential Ar should range from 10 to 100 kT . For our system with 1- μm diameter particles, the optimal zeta potential (ζ) lies between 10 and 20 mV, with the TBAB concentration around 1 μM .

Notably, the optimal ranges for asymmetric CM assembly are applicable to other systems. Based on the relationships $Ar \propto \epsilon_r\sigma_s\zeta^2$ and $\kappa \propto \sqrt{\sigma/\epsilon_r}$, achieving similar conditions in an aqueous system (where $\epsilon_r \approx 80$) would require adjustments. They include using a smaller particle size (σ_s between 200 - 500 nm), lower zeta potential (ζ between 5 - 10 mV), and increased conductivity (σ between 10 - 100 μM).

In response to the reviewer's concerns, we have explicitly clarified the role of salt concentration in tuning electrostatic interactions in both the introduction and main text. Additionally, we have expanded the Methods section to include a detailed explanation of how κ^{-1} and Q were estimated from conductivity and

zeta potential measurements. Furthermore, we have also provided guidance on selecting optimal experimental conditions based on our simulation results, emphasizing the universality of these parameter ranges. Please refer to the revised manuscript for these updates.

(2) The authors did very impressive experiments on 3D tracking of real experimental clusters as presented in Fig 1. It is clear that the angle and asymmetry can be measured for the clusters (Fig. 1d). It would be great if some of the experimental results could be compared with the extensive simulation results on the asymmetry in Fig 2. This would make it clearer to the reader how much of the simulations are verified by experiments.

Response: We apologize for not clearly explaining how we related experimental measurements to simulations in the original manuscript. In fact, we compared the configurations between experiments and simulations, and fitted the experimental parameters $\kappa\sigma_s$ and A_r . The fitted values align with the experimental trends. However, to maintain focus on the main topic, we initially chose not to include extensive details about this comparison. Since both reviewers have raised concerns on this matter, we now recognize the importance of including this comparison in the manuscript.

To match the experimental conditions with the simulation parameters, we performed simulations over a parameter space defined by two fitting variables: $\kappa\sigma_s$ and A_r . The values of A_r ranged from 5 kT to 30 kT with increments of 5 kT, while $\kappa\sigma$ varied from 2 to 16 in steps of 2. We compared the average configurations between experiments and simulations using a custom parameter δ^0 , which captures both the mean and the spread of the configurational distribution. The δ^0 parameter is defined as:

$$\delta^0 = \sqrt{\delta_m^2 + \delta_s^2}$$

where $\delta_m = \sqrt{\frac{\sum_i |\bar{r}_i^{exp} - \bar{r}_i^{sim}|^2}{N}}$ characterizes the difference between the mean

positions of vertices in experiments and simulations, $\delta_s = \sqrt{\frac{\sum_i |\Delta r_i^{exp} - \Delta r_i^{sim}|^2}{N}}$

captures the difference in positional fluctuations. Here, \bar{r}_i^{exp} and \bar{r}_i^{sim} are the mean positions of the i -th vertex in N -valence colloidal clusters, while Δr_i^{exp} and Δr_i^{sim} represent the standard deviation of the i -th vertex's position relative to their mean, from experiments and simulations, respectively. Before calculating δ^0 , we rotated the configurations from both experiments and simulations to achieve maximum overlap.

For each salt concentrations, we calculated the average δ^0 over a wide range of core particle sizes ($0.85 \sigma_s$ to $1.15 \sigma_s$) and valence numbers ($N = 2$ to 6) to identify the optimal parameter set of $(A_r, \kappa\sigma_s)$. The optimal parameters were found to be $A_r = 8 \text{ kT}$, $\kappa\sigma_s = 2$ for $1 \mu\text{M}$ TBAB, and $A_r = 5.0 \text{ kT}$, $\kappa\sigma_s = 8$ for 5

μM TBAB. Under these conditions, δ^0 was generally less than $0.1 \sigma_s$. Although the precise values differ from those estimated from experimental measurements, the trend of asymmetry indicated by our fit aligns with experimental observations. We hypothesize that this deviation may arise from the limitations of the Yukawa model at higher salt concentrations, where image charge effects become more significant.

We have now addressed this by overlaying experimental data onto the phase diagram in Fig. 2(a-c) and providing additional technical details on how we performed this comparison in the Methods section. We have also added a discussion paragraph in the main text regarding the comparison between experiments and simulations.

(3) The CLSM imaging description is in the current form not reproducible. The authors should provide information about the fluorescent dyes used (Nile – is not sufficient information). The laser excitation wavelengths and detection range, also the xy-scan speed and voxel size (or image size and volume) should be provided.

Response: We appreciate the reviewer's comment regarding the missing details in the description of the imaging technique. The PMMA particles were dyed with Nile red, excited at 552 nm, with emissions detected between 580 nm and 700 nm. The water droplets were dyed with fluorescein sodium salt, excited at 488 nm, with emissions detected between 500 nm and 540 nm.

For scanning, the system performs XY-plane scans at varying z positions. The images used for reconstruction were 128 x 128 pixels (134 nm/pixel), with a scan time of 36 ms per image, and the z-spacing between consecutive images was 250 nm. Given that the height of a typical colloidal molecule is less than 4 μm , 16 images were sufficient for complete reconstruction, with each scan taking less than 0.6 seconds. This information has been added to the revised Methods section.

It is also unclear how long the clusters were observed for the data presented in (Fig 1d).

We apologize for not clearly explaining how we obtained Fig. 1d in our original manuscript, which has caused some confusion. The distribution shown in Fig. 1d is an ensemble distribution based on snapshots from over 50 independent CMs, rather than the temporal distribution of a single CM over time. In fact, we compared the distributions from both methods and found similar results, suggesting that our system had nearly reached equilibrium by the end of the experiment. The distribution in Fig. 1d is intended to represent the equilibrium configuration.

To clarify this in the manuscript, we revised the original sentence from "Figure 1d displays the dynamic structures of CMs on a normalized sphere" to "Figure 1d shows the ensemble distribution of CMs' configurations with satellite numbers from 2 to 6, derived from the snapshot of over 50 independent CMs at equilibrium."

(4) In Fig 3c the authors present the dominant colloidal molecule N as a function of time for different Δg . The authors state that "For example, setting $\Delta g = 0.4$ kBT optimizes 4-CMs within $35 \tau_c$ ($\tau_c \equiv 1/k_0$), as depicted in Fig. 3c." However, my understanding of the data presented in Fig 3c is that 6-CMs are achieved at $t = 35 \tau_c$. Thus this claim seems to be incorrect, or the authors should explain how they came to this conclusions more clearly.

Response: We apologize for the confusion caused by the original statement. The reviewer is correct in their interpretation of Fig. 3c. At $t = 35 \tau_c$, 6-valence colloidal molecules (6-CMs) are optimized, not 4-CMs. We have corrected this error in the revised manuscript and appreciate the reviewer for bringing this to our attention.

Other small points the authors should correct:

- The authors seem to have made a very unfortunate auto-correct, they talk about cis-deCLINE in the main manuscript, this should be cis-deCALIN. The authors should correct this. (In methods it is correct).

Response: We thank the reviewer for pointing out this typo. We have corrected "cis-deCLINE" to "cis-Decalin" on page 4 of the manuscript. The main text has now been updated accordingly. Please refer to the revised manuscript for this correction.

We believe that we have adequately addressed the reviewer's comments. We hope that the reviewer finds the revised manuscript suitable for publication in *Nature Communications*.

Replies to the comments of Reviewer #4:

I have read the manuscript with interest and it contains sound, interesting and original work. However, regarding, the question whether this work is suitable for publication in NC I recommend to not publish. The main reason is that the theoretical idea of changing the ionic strength during the assembly process well defined angles can be obtained, unfortunately lacks experimental validation.

Response: We appreciate the reviewer's interest in our work and acknowledge that the experimental validation of the proposed protocol is currently limited in our manuscript. Following the reviewer's suggestion, we have made efforts to experimentally implement this pathway-control strategy, as shown in the new Fig. 4c. While tuning the ionic concentration as suggested is feasible in principle, imaging the dynamic structures and measuring the $\kappa\sigma_s$ in the emulsion droplet are technically challenging, which limits a more thorough exploration in the short term. To address the reviewer's comment, we have added a detailed description of the experimental approach used to implement the proposed method. We hope this provides readers with a clearer understanding of the experimental implementation and insights into pathway control for CM assembly. Please refer to the revised manuscript and the later response for further details.

- What are the noteworthy results?

The authors have shown interesting measurements and simulations with regards to the formation of assemblies of colloidal particles attached to an emulsion droplet. This type of process is relevant to create colloidal molecules. A challenge remains for these structures to get well defined angles between the adsorbed colloids. This paper aims to shed light on the root cause and possible resolution of this challenge. In my view the most relevant information is the measurement of the angular information of the colloids with respect to the emulsion droplet center, and the asymmetry that these angles have.

- Will the work be of significance to the field and related fields? How does it compare to the established literature? If the work is not original, please provide relevant references.

I accepted to review this paper based on the mentioning in the abstract of gradually changing the salt concentration during assembly: "We further develop a novel method to guide the ordering of CMs towards a desired structure by dynamically adjusting the ionic strength in the solvent during the ordering process." This is a nice idea but theoretically reasonably straightforward. Unfortunately, the paper does not contain experiments using gradual change in salt concentration only simulation, because of this I felt a little misguided by the abstract. I recommend stating more clearly in the abstract that at this point it is based on simulations not experiment.

Response: We appreciate the reviewer's feedback and acknowledge that our abstract may have inadvertently suggested experimental validation of the proposed approach. To clarify, we have revised the abstract to specify that the findings on kinetic control are based primarily on simulations.

While this study does not include experimental validation of the proposed protocol, it is inspired by our prior experimental observations [2]. Specifically, as shown in Fig. R2, we observed a decrease in salt concentration within the oil droplet due to ion diffusion into the surrounding water bath. This decrease is most significant in the first few hours after sample preparation, gradually slowing over time. These observations led us to consider that controlling the rate of salt concentration decrease could provide a method for tuning the kinetics of CM assembly.

Fig. R2: **a**, Schematic of the ion-diffusing process. **b**, The temporal change of $\kappa\sigma$ and structural evolution towards three different final structures: ico type (top), fcc type (middle) and bcc type (bottom). The panel a and b are extracted from the figure 1 of Ref. [2].

Fig. R3: Flow chart of an experimental plan that implements the quenching protocol.

To explore this concept, we have conducted a two-step experiment, largely following the flow chart proposed in Fig. R3. In the first step, CMs are prepared in the oil phase under a constant ionic strength. In the second step, the oil phase is combined with a water bath to form oil-in-water droplet containing CMs. Due to the ionic strength difference between oil and water phases, ions diffuse from the oil to the water, causing $\kappa\sigma$ to decrease from approximately 10 to 2 over several hours. The quench rate γ is inversely proportional to the equilibration time τ_{eq} : $\tau_{eq} \sim \Delta c V_{drop} / JS_{drop} \propto R_{drop} / D$, where Δc represents the salt concentration difference between the initial and final states of the oil droplet, J is the diffusive flux (proportional to the ion diffusion constant D), and R_{drop} is the droplet radius. By tuning the droplet size, the quench rate can be effectively controlled.

In our initial trials, we emulsified the oil-in-water droplets in the second step by vortexing rather than using microfluidics. This approach resulted in unstable droplets that tend to coalesce over time and cause inconsistent droplet sizes, leading to variations in equilibration time. To overcome these issues, a microfluidic platform capable of producing uniform and stable droplets, along with single-droplet tracking microscopy, would be ideal.

Imaging the dynamic structures and measuring the $\kappa\sigma_s$ in the emulsion droplet are technically challenging, which limits a more complete exploration in the short term. While full experimental realization is still ongoing, we believe it is valuable to include the designing of in-situ kinetic control of CM assembly. This approach provides material scientists with potential pathways for manipulating assembly dynamics. In response to the reviewer's comments, we have expanded Fig. 4 to illustrate the experimental implementation and provided additional details in the manuscript on how in-situ control over salt concentration can be achieved. These additions aim to clarify the connection between the theoretical framework and its experimental implementation.

Although the topic does contribute to a degree to the understanding of mechanisms leading to the formation of colloidal molecules having well controlled bond angles this topic is been covered in the past with more success in practical terms as such molecules have been synthesized as seen in: J.D. Kraft, J. Groenewold and W.K.Kegel, Colloidal molecules with well-controlled bond angles, *Soft Matter* 5(20), pages 3823-3826 2009

Response: We thank the reviewer for highlighting the work of Kraft et al. (*Colloidal molecules with well-controlled bond angles*, *Soft Matter*, 2009). We agree that their approach provides a practical and effective method for

synthesizing colloidal molecules with precise bond angles. By using liquid protrusions on polystyrene spheres, their technique enables the creation of colloidal structures with fixed bond angles. This precision is crucial for forming complex crystalline structures that require highly specific directional bonding.

However, the focus of our study differs significantly. While Kraft et al. excelled in synthesizing static colloidal molecules, our research centers on the **dynamic fluctuations** of colloidal molecule configurations in three dimensions. This exploration of dynamic configurations, particularly in 3D, remains largely unexplored. We believe it is crucial to study such fluctuations, especially given their relevance in biological systems. For example, the configurational flexibility in antibodies enables them to bind to antigens in various ways, allowing them to perform different biological functions [6]. Similarly, dynamic fluctuations in colloidal molecules may enhance their ability to interact with irregularly arranged binding sites, improving reaction efficiency (Fig. R4). By focusing on these fluctuations, we hope to contribute to the development of biologically inspired materials with enhanced adaptability and functionality.

Fig. R4: Schematic diagram of dynamic CMs binding

To address the reviewer's concern, we have cited Kraft et al.'s paper and other relevant works, acknowledging their contributions to colloidal molecule synthesis. Additionally, we have emphasized the distinct focus of our study and highlighted the potential insights our work can bring to the field in the Introduction.

- Does the work support the conclusions and claims, or is additional evidence needed?

There is no mention of the role of gravity on the dynamics of the particles. In my view this should be addressed, as this may bias the angles. If deemed important it should be included in the model. If not important an argument or measurement should be given to show the irrelevance of gravity.

Response: We thank the reviewer for raising this important point regarding the influence of gravity on the dynamics of colloidal molecules (CMs). While gravity may have some effect on CM configurations, we believe its impact is minimal in our system for the following reasons:

1. Estimated gravitational length significantly exceeds particle size:

In our experiments, the density of the CHB-decalin solvent is carefully adjusted to closely match the average density of the CMs ($\sim 1.18 \text{ g/cm}^3$). The density

difference between the PMMA particles and the solvent is less than 1%, while the difference between the PMMA particles and the water droplets is approximately 3%. The gravitational length h is estimated using the relationship $\frac{\pi}{6}\Delta\rho g\sigma^3 h \sim kT$, which gives $h \sim 20 \mu\text{m}$. This value is significantly larger than the particle size ($\sim 1.2 \mu\text{m}$).

2. Experimental observations:

As shown in our supplementary videos, the colloidal molecules (CMs) rotate freely without any noticeable accumulation of PMMA particles at the bottom. Additionally, a centrifugation experiment was performed at 1000 g for 5 minutes using a density-matched sample. The results showed no significant sedimentation or density variation along the Z-axis.

To address this point in the manuscript, we have updated the 'Model System Design' section to explicitly state that gravity is not considered in our model and simulations. Additionally, a more detailed justification for why gravity is not a significant factor in this system has been included in the Methods section.

As far as the modelling is concerned there is definitely an effort to model the dynamics. For the adsorption / desorption kinetics a nice validation is shown in Fig 3d and e. In Fig. 2 shows (as far as I understand) only simulations. In principle there are measurements to compare them with. Is there a specific reason why this is not shown?

Response: We appreciate the referee's valuable comment. We actually compared the configurations between simulations and experiments and used this comparison to determine the physical parameters in the experiment. Initially, we chose not to include this comparison to maintain the focus on the trends revealed by the simulations. However, we now recognize that including this comparison is important, as it strengthens the validation of our model. For further details, please refer to our response to Referee #2, Q2.

- Are there any flaws in the data analysis, interpretation and conclusions? Do these prohibit publication or require revision?

No flaws.

- Is the methodology sound? Does the work meet the expected standards in your field?

Sound methodology

- Is there enough detail provided in the methods for the work to be reproduced?

Yes

Jan Groenewold, Utrecht University

References:

- [1] Hsu, M. F., Dufresne, E. R. & Weitz, D. A. Charge stabilization in nonpolar solvents. *Langmuir* 21, 4881 – 4887 (2005).
- [2] Chen, Y. et al. Morphology selection kinetics of crystallization in a sphere. *Nature Physics* 17, 121–127 (2021).
- [3] Wodak, S. J. et al. Allostery in its many disguises: from theory to applications. *Structure* 27,566–578 (2019).

We believe that we have adequately addressed the reviewer's comments. We hope that the reviewer finds the revised manuscript suitable for publication in *Nature Communications*.

REVIEWER COMMENTS

Reviewer #2 (Remarks to the Author):

While the authors have addressed all my concerns adequately and improved the manuscript, the revised manuscript also contains changes in response to reviewer 4 which did not improve the manuscript, and I do not recommend publication in the current form.

I would like the authors to address the following issue:

Reviewer 4 has pointed out correctly that an experimental validation of the suggested pathway obtained via simulations is missing. To address this point, the authors have now added a panel (c) to figure 4. Although, I appreciate the effort of the detailed suggested preparation steps for their system, I find it does not improve the manuscript. The panel contains a lot of text, is not easy to understand and contains an extremely specified pathway of a preparation protocol not presented earlier in the paper. Hence, I find it distracts from the main message of the manuscript and does not convey the message that their findings provide guidance for an experimental pathway that would allow one to obtain controlled colloidal molecules from charged particles by simple tuning of the salt concentration during the assembly process.

I would recommend the authors to remove this specific panel (maybe add it to the supplementary information) and instead provide a more general protocol for controlled colloidal molecule preparation by specifying which particle surface charge (zeta potential) and salt concentration change rate would be needed to be achieved under certain assembly kinetics in an experiment. This would be much more impactful for the field instead of a very complex pathway that can only benefit their specific system. Especially, as it is well known that there are many different particle systems and many more ways to achieve a change in salt concentration, as has been abundantly shown for colloids. Just as an example, in a very recent work on ionic colloidal crystals an in situ change in salt concentrations was used to follow the melting process, and this simply achieved via evaporation of the solvent (Zhang et al. *Nature Materials* volume 23, pages 1131–1137 (2024)).

Seeing that the authors have provided a detailed protocol for the experiment already, I would like to ask the authors to perform the proposed experiment. I would find their findings much more impactful (and very exciting) if it was experimentally verified that their proposed method works.

Other small comment:

For figure 2 the reference to panel d and e are missing

Response: We thank the referee for their valuable and constructive comments.

We agree with the referee that performing the experiments proposed in the added panel c of Fig. 4 would strengthen the manuscript. However, as we explained in our previous letter, conducting such experiments with precise control is challenging and not straightforward. While significant efforts have already been devoted to this project, we are still working on improving the robustness of our experimental setup. As an alternative, and following the referee's suggestion, we have provided a more general and physically grounded strategy in place of the detailed experimental workflow.

To reflect this, we have moved the original Fig. 4c, which outlined the experimental implementation, to the Supplementary Information (now Fig. S4). The revised Fig. 4c now illustrates the temporal change of ionic concentration and zeta potential in a polystyrene colloidal suspension, corresponding to the ramping protocols in Fig. 4a. Additionally, we have included a paragraph explaining how to map the computed ramping protocols to physical parameters. Minor points, such as adding the missing reference in Fig. 2 and discussing alternative approaches for controlling ionic strength, have also been addressed. All changes are highlighted in blue in the updated manuscript.

We hope these revisions adequately address the referee's concerns and meet their expectations.